# Gelation in Alginate-Based Magnetic Suspensions Favored by Poor Interaction among Sodium Alginate and Embedded Particles

Alexander P. Safronov [1,2,*], Elena V. Rusinova [1], Tatiana V. Terziyan [1], Yulia S. Zemova [1], Nadezhda M. Kurilova [1], Igor. V. Beketov [1,2] and Andrey Yu. Zubarev [1]





[1] Institute of Natural Sciences and Mathematics, Ural Federal University, 19 Mira Str., 620002 Yekaterinburg, Russia; elena.rusinova@urfu.ru (E.V.R.); julja-zemova@rambler.ru (Y.S.Z.); nadyakurilova98@yandex.ru (N.M.K.); beketov@iep.uran.ru (I.V.B.); a.j.zubarev@urfu.ru (A.Y.Z.)
[2] Institute of Electrophysics UB RAS, 106 Amundsen Str., 620016 Yekaterinburg, Russia
[*] Correspondence: alexander.safronov@urfu.ru

**Featured Application: Alginate-based ferrogels and magneto-rheological suspensions are advanced materials for biomedical and bioengineering applications as scaffolds for cell proliferation, targeted drug delivery, templates for tissue regeneration and engineering.**

**Abstract:** Alginate gels are extensively tested in biomedical applications for tissue regeneration and engineering. In this regard, the modification of alginate gels and solutions with dispersed magnetic particles gives extra options to control the rheo-elastic properties both for the fluidic and gel forms of alginate. Rheological properties of magnetic suspensions based on Na-alginate water solution with embedded magnetic particles were studied with respect to the interfacial adhesion of alginate polymer to the surface of particles. Particles of magnetite ($Fe_3O_4$), metallic iron (Fe), metallic nickel (Ni), and metallic nickel with a deposited carbon layer (Ni@C) were taken into consideration. Storage modulus, loss modulus, and the shift angle between the stress and the strain were characterized by the dynamic mechanical analysis in the oscillatory mode. The intensity of molecular interactions between alginate and the surface of the particles was characterized by the enthalpy of adhesion which was determined from calorimetric measurements using a thermodynamic cycle. Strong interaction at the surface of the particles resulted in the dominance of the "fluidic" rheological properties: the prevalence of the loss modulus over the storage modulus and the high value of the shift angle. Meanwhile, poor interaction of alginate polymer with the surface of the embedded particles favored the "elastic" gel-like properties with the dominance of the storage modulus over the loss modulus and low values of the shift angle.

**Keywords:** alginates; ferrogels; magnetic nanoparticles; interfacial adhesion; dynamic mechanical analysis

## 1. Introduction

Alginate is a polysaccharide extracted from the cell walls of marine brown algae; it is widely used in various biomedical applications, healthcare, and food products. Detailed information on alginate production, properties, and applications can be found in numerous books and reviews, for instance, see references [1–5]. Polyelectrolyte behavior is the main feature distinguishing alginate from other natural polysaccharides like agarose, starch, guar, etc. Each monomeric unit in an alginate polymeric chain holds a carboxyl residue normally neutralized by a cation of a metal. If the polymeric alginate salt is formed by a monovalent cation such as Na+ or K+, alginates can dissociate in water providing highly viscous solutions. The alginates of divalent cations such as Ca++, Mg++, etc., however, are insoluble and form elastic hydrogels. In this case, the divalent cation is coordinated

by several monomeric units of adjacent macromolecules and forms a crosslink of the gel networking [1–5].

The mono/di valent ionic exchange provides the versatility of alginate applications. Thus, the fluidic water solution of sodium alginate can be readily transformed to a certain shape of an elastic gel of calcium alginate by exposure to the solution of simple calcium salts [2–6]. This feature is widely used in the encapsulation of different bioactive substances in alginate beads for their further in vivo transportation in various medical applications [7,8].

Alginate gels are extensively tested as scaffolds for cell proliferation and growth [9,10]. In these applications, the fluid-to-gel transition of alginate driven by the mono/di valent ionic exchange is especially useful for the in vivo tissue regeneration and engineering [11]. Fluidic sodium alginate solution can easily be injected into a living organism and the gelation might then be provided in the desired place of the interior. In this regard, the modification of alginate gels and solutions with dispersed magnetic particles gives extra options to control the rheo-elastic properties both for the fluidic and gel forms of alginate [12]. Several studies on the structure and magneto-rheological properties of alginate ferrogels had been reported [13–18]. Thus [14,15], ferrogels filled with iron micron-sized magnetic particles were prepared by the $Na+/Ca++$ ionic exchange in magnetic suspensions in alginate solution according to the "internal gelation" procedure, and their mechanical properties were tested using dynamic mechanical analysis (DMA). It was shown that Ca-alginate ferrogels had substantially higher levels of the storage (elastic) modulus compared to the loss modulus within the linear viscoelastic region, which is the typical behavior of the cross-linked polymeric networks. The values of viscoelastic moduli intensively enhanced if the concentration of magnetic particles increased by up to 30% (vol). The application of the magnetic field with up to 300 $kA/v$ intensity considerably strengthened the ferrogels: for instance, the storage modulus increased by seven-fold approximately. DMA study was also performed on Ca-alginate ferrogels with rod-like composite particles with magnetite shells [17] and the same magnetorheological effect for the influence of the applied magnetic field on the viscoelastic moduli was revealed. The enforcement of the gel network by the embedded particles was reported for Ca-alginate ferrogels with super-paramagnetic iron oxide particles [16].

The properties and behavior of alginate ferrogels are the implicit functions of their internal structure which is governed by multiple types of molecular non-valent interactions among the constituents, which include the polymer network, the particles, and the water medium. Whereas the interaction of alginate with water might be considered the same for all alginate-based ferrogels, the interaction between alginate and embedded magnetic particles might be different depending on the chemical nature of the particles. It puts forward a question of whether interaction among particles and alginate would notably affect the viscoelastic properties of alginate ferrogels. In the present study, we have examined this opportunity for the viscoelastic suspensions of magnetic particles in sodium alginate solutions.

We have taken four different types of submicron-sized magnetic particles: magnetite, metallic iron, metallic nickel, and metallic nickel with deposited carbon shell and used them for the preparation of magneto-rheological suspensions (MRS) based on sodium alginate solution in water. Viscoelastic moduli were determined for these suspensions as a function of the content of the embedded magnetic particles. Using the thermodynamic approach based on experimental calorimetry studies we have evaluated the enthalpy of interaction between alginate polymer and these types of magnetic particles. It will be shown below that interaction between alginate and particles strongly affects the rheological behavior of the suspension and can even cause its gelation.



## 2. Materials and Methods

### 2.1. Magnetic Particles

Sodium alginate was a commercial product from Sigma-Aldrich, Merck (St. Louis, MO, USA). The molecular weight of the polymer was determined by viscometry in 0.1 M NaCl. It was found 190 kDa using Mark-Houwink constants K = 0.023 cm$^3$/g, a = 0.984 [19].

Magnetite (Fe$_3$O$_4$) was a commercial product from Alfa Aesar (Alfa Aesar, Ward Hill, MA, USA). Phase composition determined by X-ray diffraction (XRD) was: Fe$_3$O$_4$ phase— 94% wt., Fe$_2$O$_3$ phase—1% wt., and FeO(OH) phase—5% wt. TEM microphotographs of magnetite particles are given in Figure 1a. The particles were quasi-spherical with a caliper diameter in the range of 50–500 nm. Weight average caliper diameter (D$_w$) determined via particle size distribution (PSD) from the image analysis of 632 particles was found 305 nm. The saturation magnetization of magnetite particles was 84 emu/g, remnant magnetization was 6.6 emu/g, and coercivity was 78 Oe (Table 1).

Metallic iron (Fe), metallic nickel (Ni), and nickel coated with carbon shell (Ni@C) were synthesized by the electrical explosion of wire (EEW) method, which technical details and the design of the laboratory setup were given elsewhere [20,21]. In brief, the EEW method to produce metal particles is based on the evaporation of a portion of a metal wire by the high voltage electrical discharge in an inert working gas (argon) and further condensation of particles from vapor. In the case of Ni@C particles butane was added to the working gas. It was decomposed in the electrical discharge arc and gave carbon vapors, which were deposited onto the surface of metallic Ni. The details of the deposition setup are given in earlier works [22,23]. TEM microphotographs of Fe, Ni, and Ni@C particles are given in Figure 1b–d. The particles had a spherical shape and their diameter was in the range of 10–200 nm. The weight average diameters of these batches of particles (Table 1) were determined from their PSDs obtained by the image analysis of microphotographs. The phase composition of the metallic Fe and Ni particles and of the metallic core for Ni@C particles was 100% cubic metal phase as determined via XRD. Selected characteristics of magnetic particles are given in Table 1.

### 2.2. Suspensions and Composites

To prepare magnetic suspensions first the stock solutions were made by dissolving sodium alginate in distilled water under permanent stirring. Weighted portions of magnetic particles were dispersed in a small amount of water under ultrasound treatment for 5 min, and then mechanically homogenized with viscous alginate solution in a rotational dissolver at 2000 rpm for 30 min. Resulted suspensions were kept overnight for equilibration. As the sterile conditions were not specially maintained, the rheological testing was performed within 36 h after sodium alginate was dissolved to prevent biodegradation of alginate in solutions and in the suspensions.

Separately, binary polymeric compositions of alginate with different batches of magnetic particles were prepared for thermodynamic measurements. Therefore, magnetic particles were dispersed in a small amount of water under ultra-sound treatment and the resulting magnetic slurry was then mixed with a 7% stock solution of alginate in a rotational dissolver at 2000 rpm for 20 min. Certain proportions among magnetic slurry and alginate solution were kept maintaining the series of suspensions in which the polymer-to-particle weight ratio varied discretely. Resulted suspensions were then cast onto the glass surface and water was evaporated at 50 °C to the constant weight of the dry polymeric film. Thus, polymeric films of magnetic compositions were prepared with sodium alginate content ranging from 0 to 100% with a 10% step.

**Table 1.** Selected properties of magnetic particles used in alginate-based magneto-rheological suspensions.

| Batch | $S_{sp}$ (m$^2$/g) * | $D_w$ (nm) | $M_0$ (emu/g) | $M_r$ (emu/g) | $H_c$ (Oe) |
|---|---|---|---|---|---|
| Fe$_3$O$_4$ | 6.9 | 305 | 84 | 6.6 | 78 |
| Fe | 9.0 | 86 | 154 | 14.9 | 280 |
| Ni | 12.6 | 85 | 51 | 17.7 | 260 |
| Ni@C | 14.2 | 98 | 47 | 6.9 | 128 |

* Specific surface area as determined via low-temperature sorption of nitrogen.

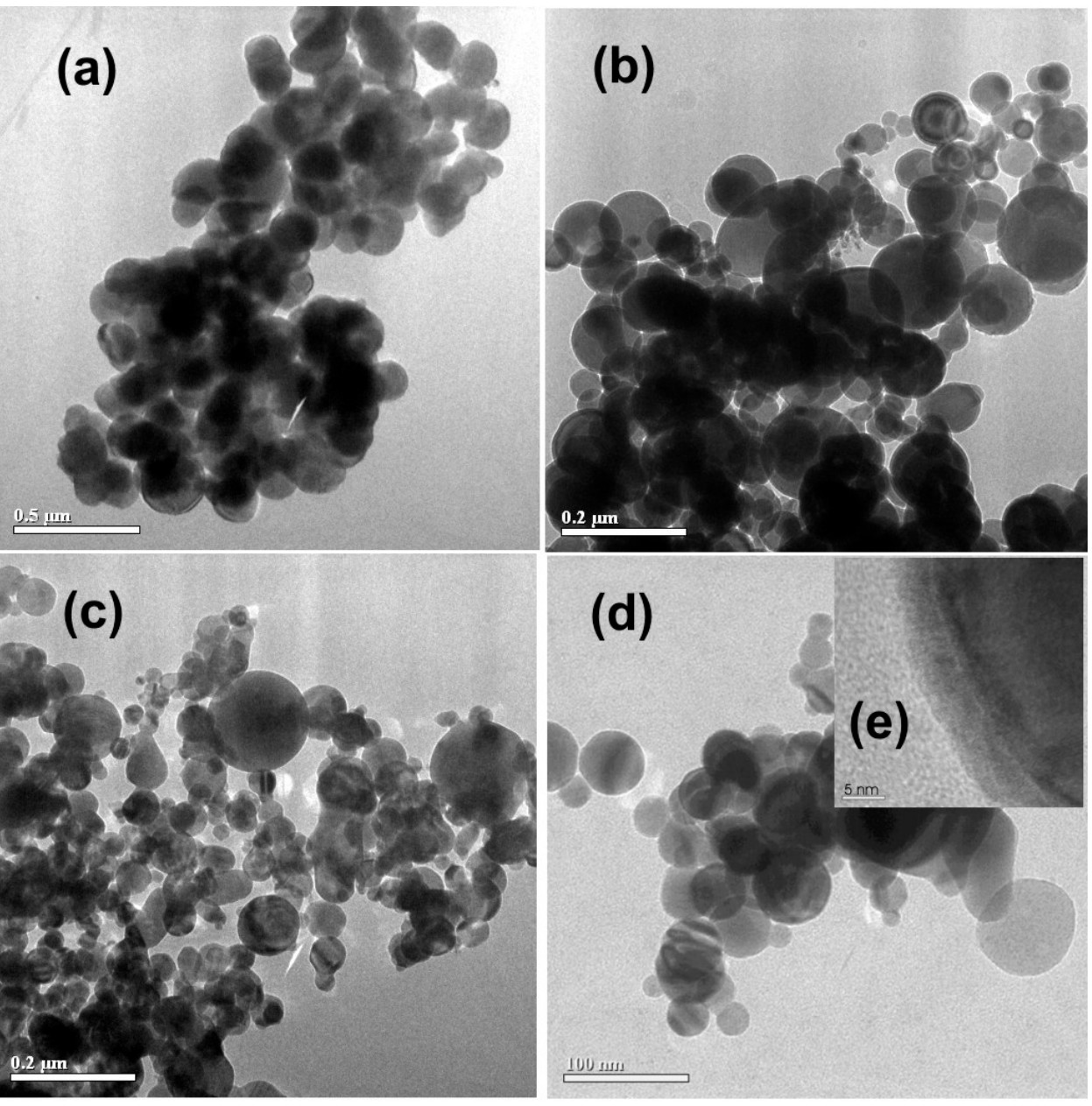

**Figure 1.** TEM microphotographs of magnetic particles: (**a**)—magnetite (Fe$_3$O$_4$); (**b**)—metallic iron (Fe); (**c**)—metallic nickel (Ni); (**d**)—nickel particles coated with carbon shell (Ni@C); (**e**)—HRTEM image of the carbon layer on the surface of Ni particle.

*2.3. Methods*

TEM characterization of magnetic particles was conducted using a JEOL JEM2100 microscope operated at 200 kV (JEOL, Tokyo, Japan). Phase composition was examined using an X-ray diffractometer Bruker D8 Discover (Bruker Corporation, Billerica, MA, USA) operated at Cu $K_a$ radiation (wavelength $\lambda$ = 1.5418 A) with a graphite monochromator and a scintillation detector. The specific surface area ($S_{sp}$) of particles was measured via low-temperature adsorption of nitrogen by Brunauer-Emett-Teller calculation procedure using Micromeritics TriStar3000 analyzer (Micromeritics, Norcross, GA, USA). The magnetic hysteresis loops were measured at room temperature by a vibrating sample magnetometer (VSM Cryogenics Ltd., London, UK). Viscoelastic properties of magneto-rheological suspensions were characterized by the dynamic mechanical analysis (DMA) using Haake Mars rheometer (ThermoFisher Scientific, Waltham, MA, USA) in the oscillatory mode. The relative error in the determination of the storage modulus and the loss modulus of the suspensions was less than 5%. The suspensions were examined as droplets put in a standard cone-and-plate cell of the rheometer. Hence, no definite geometry of a sample was provided. At a high load of magnetic particles, we have, however, found that the behavior of the liquid suspensions tended to gelation and the suspensions showed up increasing elasticity. Nevertheless, they still retained restricted fluidity and we were able to keep the initial cone-and-plate setup of the rheometer.

The enthalpy of interaction among alginate polymer and magnetic particles was determined using a thermochemical cycle which is presented below in the Results section. This cycle elaborates the values of the enthalpies of dissolution of alginate composites with embedded magnetic particles. They were measured using a 3D Calvet calorimeter SETARAM C80 (SETARAM, Caluire, Auvergne, France) at 25 °C. A pre-weighted amount of a composite film was placed in a thin glass ampoule, which then was sealed with a burner. A total of 7 mL of water was placed in a special holder into a stainless-steel calorimetric cell. The assembled cell was mounted in the calorimeter and thermally equilibrated overnight. The experiment started by breaking the ampoule in water inside the cell using a breaking rod of an ampoule vessel. Heat evolution curve was then recorded. The typical experiment lasted 2 h until the initial baseline was re-established. Integration of the heat evolution curve was conducted using the CALISTO software, which gave the value of the enthalpy of dissolution then used in the thermochemical cycle. The absolute error of the measurement of enthalpy of dissolution was less than ±0.02 J.

**3. Results**

As the first step in the dynamic mechanical analysis of alginate-based magneto-rheological suspensions, we tested water solutions of sodium alginate in concentrations 5, 10, and 15% (wt.). Frequency dependence experiments in the frequency range 0.1–100 Hz at 5 Pa stress showed that $G'$ and $G''$ were linearly increasing in double log scale, with $G''$ > $G'$ and their cross-over at approximately 80 Hz. Based on these results, the frequency of 1 Hz was chosen as the level for the study on stress-dependent rheological properties for MRSs. Typically, the process of gelation in polymer solutions is examined at a low frequency of the oscillatory stress and we took 1 Hz as a close value to a lower limit in frequency measurements (0.1 Hz). Figure 2 presents the shear stress dependence of storage modulus ($G'$), loss modulus ($G''$), and shift angle ($\delta$) between the applied stress and the strain in DMA measurement at the frequency of oscillations 1 Hz for two concentrations of Na-alginate (5 and 10%). The dependences for the 15% solution were very much like that for the 10% solution but absolute values of moduli were higher by approximately an order of magnitude.

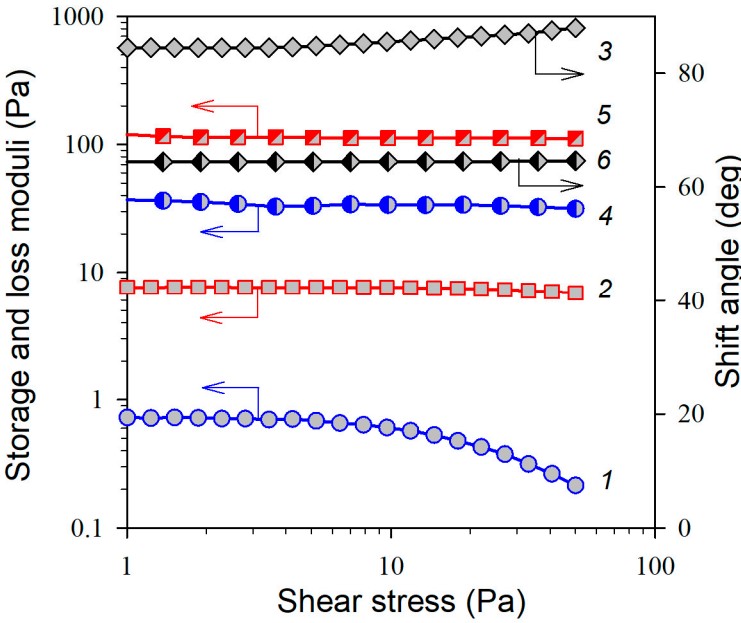

**Figure 2.** Shear stress dependence of the dynamic mechanical parameters for Na-alginate water solutions. Frequency of shear oscillations 1Hz, with a temperature of 25 °C. Concentration 5% (wt.): 1—storage modulus, $G'$; 2—loss modulus, $G''$; 3—shift angle, $\delta$. Concentration 10% (wt.): 4—storage modulus, $G'$; 5—loss modulus, $G''$; 6—shift angle, $\delta$.

In the range of shear stress, 1–50 Pa the values of dynamic moduli were constant for both concentrations, except the storage modulus for 5% solution. In that case, $G'$ kept constant in the range of shear stress 1–8 Pa and gradually diminished at higher values. According to common settlement [24] the plateau on the shear stress dependence of modulus is an intrinsic feature of the gel-like behavior of polymeric systems. In the case of Na-alginate it, in general, indicated the presence of physical gelation due to the formation of H-bonds.

Meanwhile, for all concentrations of Na-alginate, the values of loss modulus ($G''$) were substantially higher than the values of storage modulus ($G'$). It is clear from Figure 3, which presents the dependences of $G'$, $G''$, $\delta$ on Na-alginate concentration at 1 Hz frequency and selected shear stress $\tau = 1$ Pa. The values at 1 Pa were taken as characteristics for the modulus plateau.

The prevalence of $G''$ over $G'$ means that the balance in the viscoelastic behavior of the Na-alginate solution is biased to fluidic properties. It is clearly indicated by the value of the shift angle. For the idealized elastic material, the strain is in-phase with the stress and the shift angle is 0, while in the idealized fluid, it is 90 degrees. The shift angle was 85 degrees for the solution with 5% Na-alginate and it indicated that the viscous flow dominated the dynamic deformation. As the shift angle went down to 72 degrees for 10% solution and to 57 degrees for 15% solution, it meant that the elastic contribution to deformation progressively increased with the concentration of Na-alginate. It looks reasonable, as the upsurge in concentration had to result in the increase of the probability of inter-chain contacts, which were the origin of physical networking due to H-bonding.

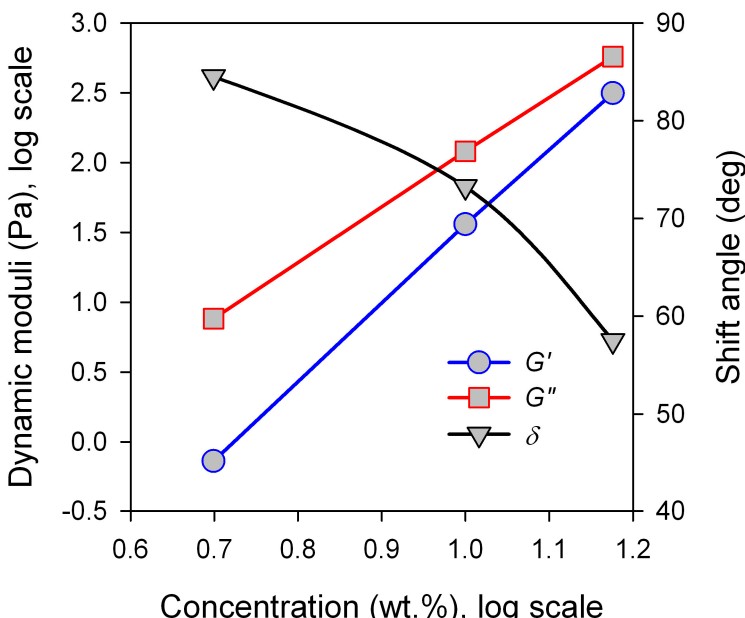

**Figure 3.** Concentration dependences of the dynamic moduli and shift angle at 1 Hz frequency and shear stress 1 Pa.

The dependences for $G'$ and $G''$ were found linear in the double-logarithmic scale. It means that $G'$ and $G''$ had a power dependence on the concentration of Na-alginate in water ($C$). It led to the following scaling equations:

$$G' \propto C^{5.5}$$
$$G'' \propto C^{3.9}$$
$$(1)$$

The power exponents in Equation (1) are substantially larger than the classical exponent 1/3 in Flory-Rehner theory for the elasticity of polymeric networks [25]. Most likely it is the consequence of the dominance of the fluidity of the solutions. Although the elastic contribution to the deformation of Na-alginate solutions is clearly present, it is still less pronounced than the contribution from the viscous flow. In other words, the solutions of Na-alginate are more "liquids" than "elastomers". At the molecular level, it probably means that the H-bonds, which presumably act as the cross-links in the physical networking of the solution, are unstable and they constantly appear and dissolve in a stochastic fashion. It is also likely that the polyelectrolyte nature of Na-alginate contributes considerably to the fluidity of the solution. Basically, interaction among negatively charged ionized base-units of the Na-alginate chain is repulsive and it hampers the formation of stable cross-links in the physical network of inter-chain H-bonds.

Let us now turn to the dynamics of magneto-rheological suspensions (MRS) based on Na-alginate solutions with dispersed magnetic particles (MPs). For technical reasons, MRS were prepared using a 5% solution of Na-alginate only. The viscosity of 10% and 15% solutions was too high to provide a uniform mixing with MNPs.

Figure 4 presents the shear stress dependences of storage modulus ($G'$), loss modulus ($G''$), and shift angle ($\delta$) at 1 Hz for MRS based on 5% Na-alginate solution with 2.8% (wt.) of $Fe_3O_4$ MPs. The dependence of all these dynamic characteristics was essentially the same as that presented in Figure 2 for the 5% Na-alginate solution without MPs. The storage modulus had a plateau at a low level of shear stress and moderately diminished with its increase. The loss modulus preserved an almost constant value and was substantially higher than the storage modulus. The shift angle was almost constant, and it was close to 90 degrees. It all meant that MRS with 2.8% (wt.) of $Fe_3O_4$ MPs demonstrated the same liquid-type behavior as the carrier solution of Na-alginate.

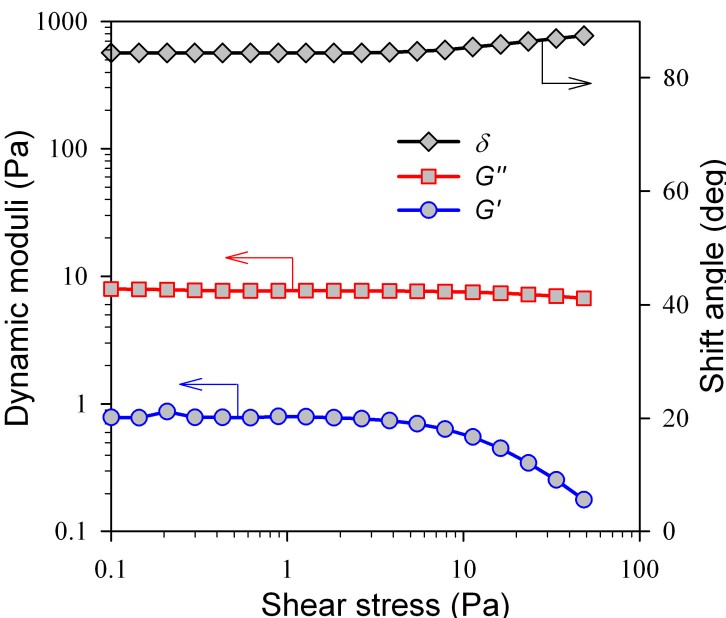

**Figure 4.** Shear stress dependence of the storage modulus ($G'$), the loss modulus ($G''$), and the shift angle ($\delta$) for the magneto-rheological suspension of 2.8% $Fe_3O_4$ MPs dispersed in 5% Na-alginate solution. Frequency of shear oscillations 1 Hz, temperature 25 °C.

Similar features were observed for suspensions with all other MPs, which contained 2.8% (wt.) of MPs. The dependences of $G'$, $G''$, $\delta$ were qualitatively very much like that for the initial 5% Na-alginate solution. However, the situation changed drastically if the content of MPs in suspensions was enlarged.

Figure 5 presents the dependence of the dynamic moduli ($G'$ and $G''$) on the volume fraction of MPs in suspension. The values of $G'$ and $G''$ were averaged over the range of their constancy, which was 0.1–1 Pa.

In the case of $Fe_3O_4$ MPs, both dynamic moduli moderately increased with particle volume fraction. At any concentration, the loss modulus was higher than the storage modulus. In the case of Fe, Ni, and Ni@C MPs there was a steep increase in both moduli if the content of particles exceeded 3–4% (vol.) Note that the ordinate in Figure 5 is logarithmic and the growth in the moduli was several orders of magnitude. For instance, the storage modulus of 12% (1.7% vol.) suspension of Fe MPs was $G' = 3.1$ Pa and for 30% (5.2% vol.) it had lifted to 42.3 kPa, which was exactly 4 orders of magnitude boost. According to the degree of elevation of $G'$ with concentration the MPs fell in a sequence: $Fe_3O_4$–Ni@C–Ni–Fe.

The loss modulus of suspensions also enlarged with MPs concentration. The influence of the nature of MPs on the degree of elevation was the same as for $G'$. At the same time the increase in $G''$ was not as drastic as the increase in $G'$. As a result, there was a cross-over in the $G'$ vs. $G''$ relation at high content of Ni@C, Ni, Fe MPs in the suspension. While at low concentrations of these MPs $G''$ was larger than $G'$, at high concentration this relation had reversed to $G'' < G'$. It meant that at high MPs content (except $Fe_3O_4$ MPs) elastic deformation prevailed over the viscous dissipation of mechanical energy.

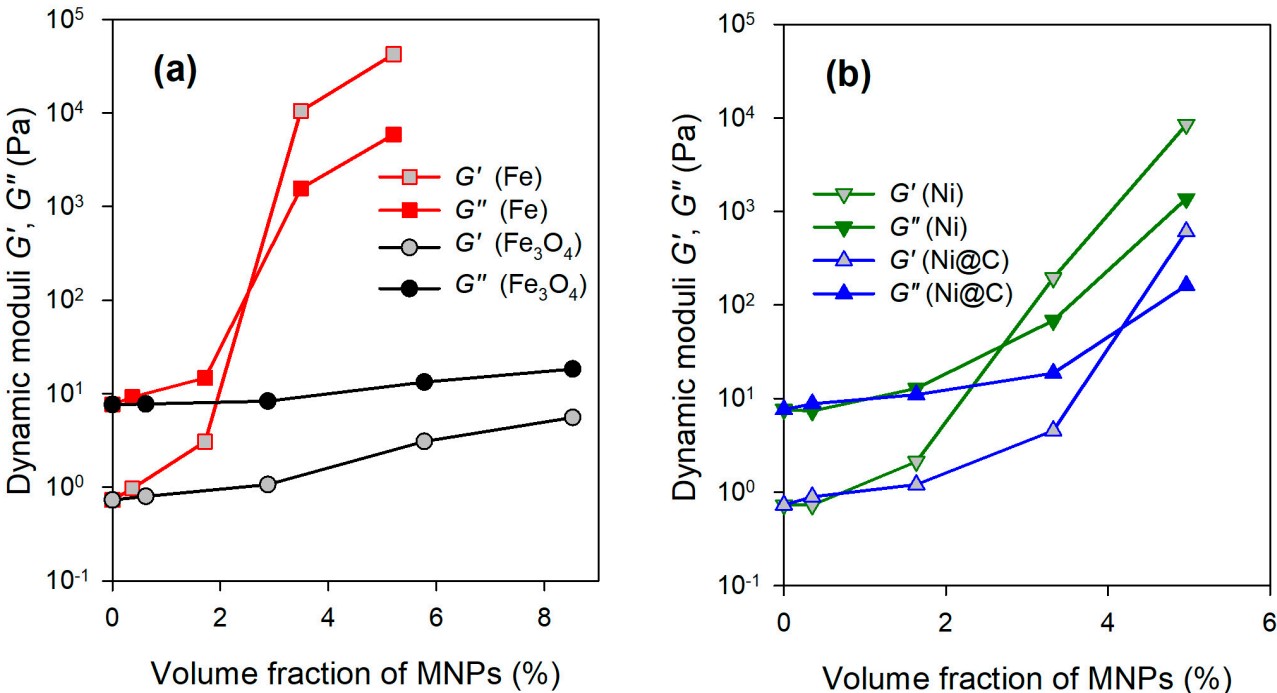

**Figure 5.** Concentration dependences of storage modulus $G'$ and loss modulus $G''$ for the magneto-rheological suspensions with different magnetic particles. (**a**)—Fe and $Fe_3O_4$ MPs, (**b**)—Ni and Ni@C MPs. $G'$ and $G''$ are averaged over the plateau in 0.1–1 Pa range of shear stress. Frequency 1 Hz. Temperature 25 °C.

It can be illustrated by Figure 6 which gives the dependence of the shift angle on MPs concentration in the suspensions.

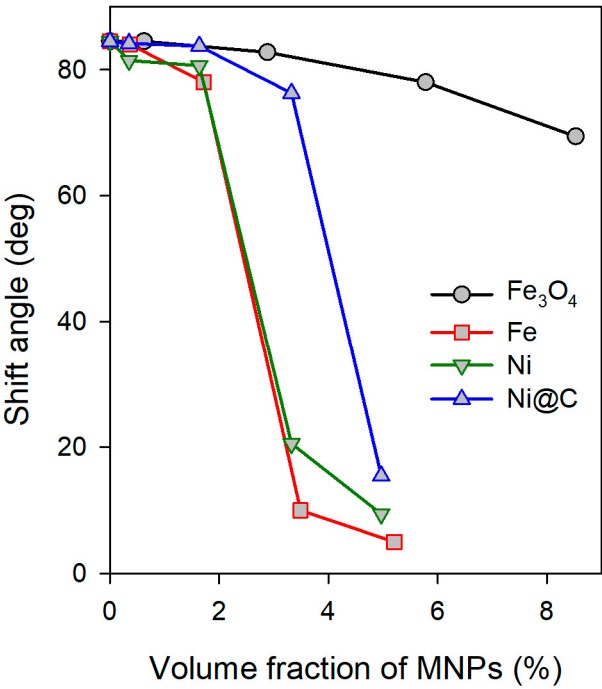

**Figure 6.** Concentration dependences of shift angle $\delta$ for magneto-rheological suspensions with different magnetic particles. $\delta$ is averaged over the plateau at 0.1–1 Pa range of shear stress. Frequency 1 Hz. Temperature 25 °C.

It is clear from Figure 6 that at low concentrations of MPs the dynamic deformation of suspensions was predominantly the viscous flow, as the shift angle was above 80 degrees close to the value of 90 degrees for the ideal fluid. In the case of suspensions with $Fe_3O_4$ the fluidic nature of deformation preserved up to the highest content of MPs (30% wt., 8.3% vol.). Meanwhile, for the suspensions with other MPs, the turn-over occurred for the high concentration of MPs. The shift angle for the suspensions with Ni@C, Ni, and Fe MPs dropped down to values below 10 degrees which were close to the characteristic shift angle $\delta = 0$ for the ideal solid. It meant that magneto-rheological suspensions with Ni@C, Ni, and Fe MPs in a concentration above 3–4% (vol.) became elastic gels rather than viscous fluids. Note that even in the fluidic state the effect of the particle concentration on the moduli is much higher than that predicted by the classical Einstein theory of effective viscosity of suspensions.

According to the degree of the descent of the shift angle, the suspensions form a sequence $Fe_3O_4$–Ni@C–Ni–Fe, which is identical to that marked above concerning $G'$ data.

What is the underlying reason for the transition of fluid to elastic gel to occur in the case of suspensions with Ni@C, Ni, and Fe MPs, and not to happen in the case of suspension with $Fe_3O_4$? Our hypothesis was that its background is in different interfacial interactions in these suspensions. To check this supposition, we studied the interfacial adhesion of Na-alginate polymer to the surface of MPs using the thermodynamic approach, which was successfully elaborated for this purpose in earlier studies [26–28].

From the viewpoint of basic thermodynamics, the intensity of the adhesion of polymer at the interface with solid might be characterized by the value of the enthalpy of adhesion ($\Delta H_{adh}$), which is the gain in enthalpy if the polymer is put in contact with a solid surface. The thermodynamic equation for this process is:

$$\text{Na-alginate polymer + solid MNP = Composite} + \Delta H_{adh} \qquad (2)$$

The heat effect of this process cannot, certainly, be measured directly as both interacting substances are solids. In this case the appropriate thermodynamic cycle is elaborated, which includes the sequence of measurable steps from the initial state to the final state. In the case of composite formation, these steps are based on the dissolution of polymer and composite in a good solvent. Then, water taken as a good solvent for Na-alginate and its composites, the steps of the thermodynamic cycle which corresponds to Equation (2) are:

$$\text{Na-alginate polymer + water = Solution} + \Delta H_1 \qquad (3)$$

$$\text{Solid MNPs + water = Suspension 1} + \Delta H_2 \qquad (4)$$

$$\text{Solution + Suspension 1 = Suspension 2} + \Delta H_3 \qquad (5)$$

$$\text{Composite + water = Suspension 2} + \Delta H_4 \qquad (6)$$

All these steps can be performed in a calorimeter and the resulting combination of the enthalpy values reads:

$$\Delta H_{adh} = \Delta H_1 + \Delta H_2 + \Delta H_3 - \Delta H_4 \qquad (7)$$

It is worthwhile noting that values $\Delta H_1$ and $\Delta H_2$ relate to individual components but the value $\Delta H_4$ for the dissolution of a composite is a function of the polymer content in it. Typically, the value $\Delta H_3$ related to the mixing of a solution with a suspension of MPs (Equation (5)) is small, it falls within the limits of the experimental error for the other values in the cycle, and it is usually neglected.

Composites based on Na-alginate polymer with embedded MPs in different content were prepared separately as described above in the Materials and Methods part. Figure 7 presents the concentration dependence of the enthalpy of dissolution ($\Delta H_4$) of these com-

posites in water at 25 °C measured in a direct experiment in calorimeter. Ni and Ni@C MPs were taken as examples.

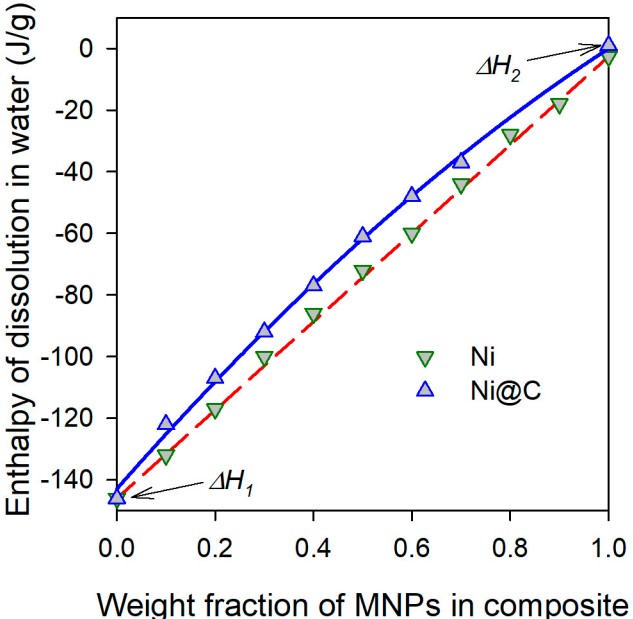

**Figure 7.** Enthalpy of dissolution of Na-alginate + MPs composites in water ($\Delta H_4$) at 25 °C. Values at the ordinate axes correspond to the enthalpy of dissolution of Na-alginate (left axis—$\Delta H_1$) and to the enthalpy of wetting of MNPs (right axis—$\Delta H_2$). Experimental data for Ni and Ni@C are presented.

Figure 7 contains all the necessary information to calculate the enthalpy of adhesion using the thermodynamic cycle in Equation (7). The value at the left-side ordinate axis corresponds to the composite with 0% content of MPs, which is pure Na-alginate. It is the $\Delta H_1$ value in Equation (3). The value at the right-side axis is $\Delta H_2$ in Equation (4) as it corresponds to the composite with 100% of MPs. Other points in the plot correspond to the values of the enthalpy of dissolution of composites with progressively increasing MPs content ($\Delta H_4$ in Equation (6)).

In fact, any composite is a mixture of components and even if there is no interaction between components, the enthalpy of dissolution would depend on the proportion of the components in a mixture as the values of $\Delta H_1$ and $\Delta H_2$ are different. Theoretically, the enthalpy of dissolution of an additive mixture of components would be a linear function between the values for the constituents. It is noticeable in Figure 7 that it was the case of Na-alginate composites with embedded Ni MPs. The experimental points for the enthalpy of dissolution in this system are close to a straight line between $\Delta H_1$ and $\Delta H_2$. Meanwhile, the values of the enthalpy of dissolution for Na-alginate composites with Ni@C deviated upward from the straight line of an additive mixture. It meant that there was the interaction between components in composites with Ni@C particles, unlike the system with Ni MPs. Most likely it was the result of the surface modification of Ni with a carbon layer.

Experimental data presented in Figure 7 and similar data for Na-alginate composites with $Fe_3O_4$ and Fe MPs were used for the calculation of the enthalpy of adhesion of Na-alginate to the surface of MPs using Equation (7). The results are plotted in Figure 8, which gives concentration dependence on the enthalpy of adhesion. The values of the enthalpy of adhesion are related to the unit area ($m^2$) of the surface of the embedded MPs.

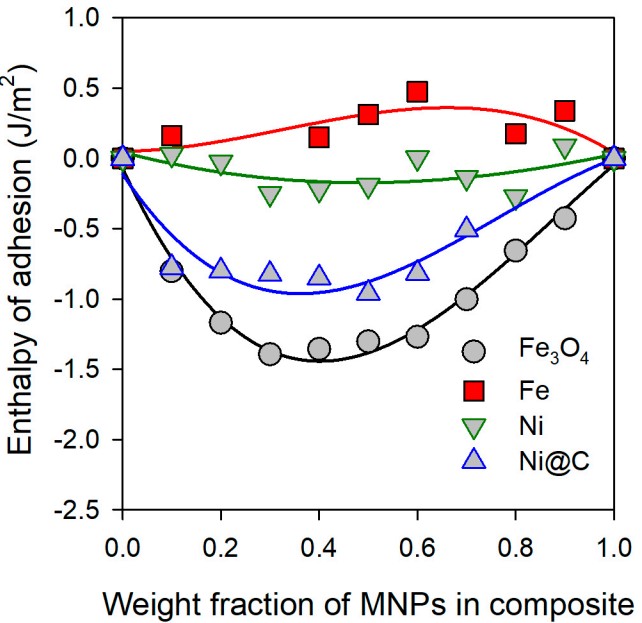

**Figure 8.** Concentration dependences of the enthalpy of adhesion of Na-alginate polymer to the surface of MNPs. Temperature 25 °C. Lines are drawn for the eye-guide only.

The dependences in Figure 8 are cupola curves, which pass through zero at both ends of the abscissa axis, that is at 0% and 100% of MPs content. The values of $\Delta H_{adh}$ were negative in the case of Na-alginate composites with $Fe_3O_4$ and Ni@C composites, they were very close to zero for the composites with Ni and were slightly positive for the composites with Fe. It meant that the interaction of Na-alginate with $Fe_3O_4$ and Ni@C MPs is energetically favorable and there is a gain in enthalpy in this process. There is almost no interaction of Na-alginate with Ni MPs, and interaction with Fe MPs is slightly unfavorable.

It is noticeable that the values of the enthalpy of adhesion for Na-alginate increase in a sequence $Fe_3O_4$–Ni@C–Ni–Fe, which is the same sequence as that pointed out above concerning $G'$, $G''$, and $\delta$ for magneto-rheological suspensions. Combining these two results we may conclude that good interaction of Na-alginate with $Fe_3O_4$ MPs resulted in fluid-type behavior of suspensions (see Figure 6). On the contrary, Fe MPs poorly interacted with Na-alginate, the enthalpy of adhesion was positive, and the suspensions with Fe MPs exhibited dominance of elasticity at their volume fraction above 3% (vol.).

The pair Ni–Ni@C showed the same feature. Ni MPs were almost non-interacting with Na-alginate and it resulted in the distinct elasticity of suspensions with a volume fraction of Ni above 3% (see Figure 6). Deposition of the carbon layer had improved the adhesion of Na-alginate to the surface of MPs but it resulted in a drawback in the elasticity of the suspensions. As one may see in Figure 6, the suspensions with 3.5% (vol.) of Ni@C MPs remained fluids, while the suspensions with Ni MPs at the same volume fraction were dominantly elastic.

## 4. Discussion

From the basic physical point of view, any measurable property of the physicochemical system takes its origin in the balance of all molecular interactions of the interior. If there are no chemical reactions in the system as in the case of alginate-based MRSs then all the features of the macroscopic properties are governed by the following types of interaction: hydration of alginate, hydration of solid magnetic particle, interfacial adhesion of alginate to the surface of particles, and magnetic forces among particles. The study was focused on the influence of interfacial adhesion on the rheological properties of alginate MRSs. Presented experimental results revealed the correlation between the viscoelastic properties of alginate-based suspensions and the enthalpy of adhesion of alginate polymer to the

surface of magnetic particles in composites. If the values of the enthalpy of adhesion of Na-alginate to the solid surface of MPs were negative, which meant the energetically favorable interaction, then it led to a fluidic type of deformation of magneto-rheological suspensions at the same time. In other words, good interfacial adhesion at the interface depressed the elevation of the dynamic moduli and the descent of the shift angle with the increase in MPs content. Thus, the enhancement of polymer-solid interaction might be favorable to preserve the fluidity of the suspension. Oppositely, if the interaction of the polymer and the solid at the interface was poor, it provided the boost of the dynamical moduli and the dropdown of the shift angle at a relatively low volume fraction of MPs. Thus, based on the obtained results we may conclude that to shift the properties of MRS to elasticity rather than fluidity and to transform the fluid to a gel phase one needs to block polymer-solid interaction somehow.

At the molecular level, we may suppose the following explanation of the observed dependence of the DMA behavior of MRS on the interaction at the interfaces. If the interaction of Na-alginate with the surface of MPs is energetically favorable, then polymer chains adsorb at the surface and form a steric protective layer which prevents a close contact between magnetic particles. Based on the extended Derjaguin-Landau-Verwey-Overbeek theory it was shown earlier [26], that magnetic interactions between Fe MPs in colloids dominated over van der Waals and electrostatic forces. Therefore, thick steric protective polymeric layers on the surface of MPs are needed to prevent aggregation of MPs due to the magnetic forces and to provide colloidal stability of the suspension.

In the present study, no special alignment of particles in the preparation of MRSs was provided. Therefore, the distribution of magnetic particles in MRS was macroscopically random. Meanwhile, the possibility of aggregation at the microscopic scale could not be completely excluded. The analysis of particle aggregation in condensed phases is very complicated and to a large extent an unresolved problem. In the present study, we could not evaluate the aggregation as the content of magnetic particles in suspensions was too large for optical microscopy. In this respect, the following model consideration might refer not only to individual MPs but too small aggregates of MPs as well.

The caliper diameter of MPs in the present study (see Table 1) was above the threshold to a single domain state which is about 128 nm for $Fe_3O_4$, 14–30 nm for Fe, and 55 nm for Ni [29]. Thus, the MPs under study consisted of several domains each. In this situation magnetic interaction between the nearest domains in the closely situated particle can be not fully compensated by the interaction between the other domains in these particles and the total magnetic interaction of these "bare" particles can be quite significant, provoking their agglomeration and gelation of the suspension. The ensemble of individual MPs covered with the polymeric protective layer and with partially compensated domain magnetic moments would likely favor the independent mobility of particles under the applied mechanical stress and consequently the fluid-type deformation. Schematically this scenario is given in Figure 9a. Apparently, it occurred in the case of $Fe_3O_4$ and Ni@C MPs.

Figure 9b schematically illustrates the other situation if there is no interfacial adhesion of Na-alginate at the surface of MPs. In this case, polymer does not adsorb at the surface and there is no polymeric layer at the surface which prevents close contact between particles. At low MPs concentration if the probability of inter-particle contacts is small this case does not differ from the previous one. The situation changes if the content of particles increases, and the probability of their contact becomes substantial. As MPs are not covered with steric polymeric layers, they may establish close contact and aggregate with each other. In this aggregation, magnetic forces may appear between domains that are located at the neighboring particles. Although the cumulative magnetic moment of the multi-domain particle is close to zero, domains at the surface have non-zero moments. Then, the magnetic interaction of local magnetic moments at the surfaces of neighboring particles may take place at their close contact. As a result of this interaction, a network structuring of magnetic particles may appear, which causes the gelation of the suspension. The evidence in favor of this scenario was reported in reference [30] for ferrocolloids with Co nanoparticles, which

formed chain-like structures, and in reference [31] concerning ferrogels with embedded strontium hexaferrite magnetic particles which showed network-type aggregation of multi-domain magnetic particles. The networking of magnetic particles due to their orientation by attractive magnetic interaction apparently would lead to the prevalence of the elastic deformation of MRS.

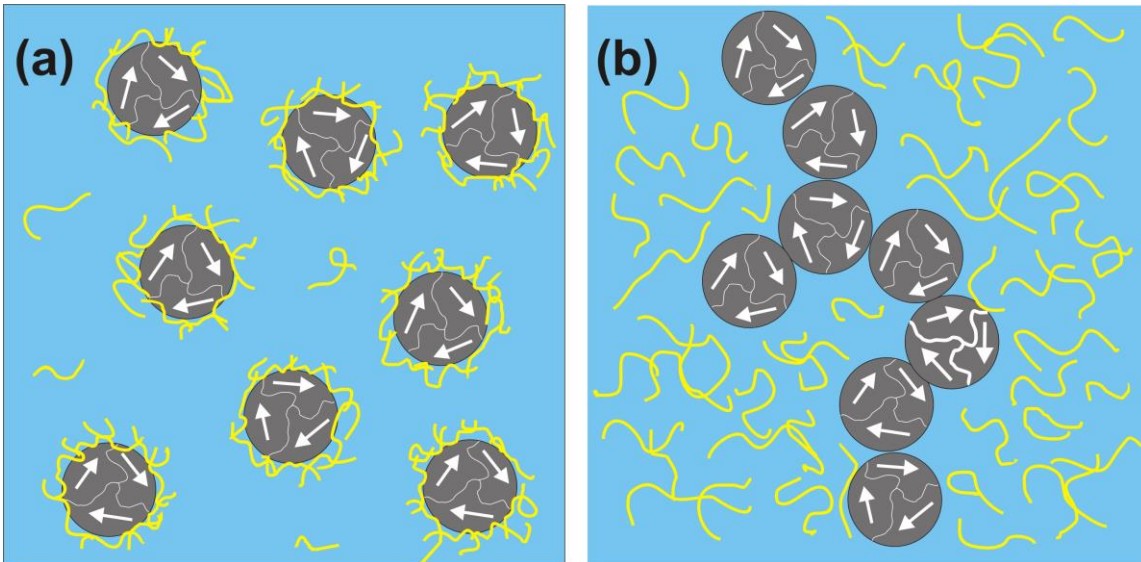

**Figure 9.** Schematic presentation of the two types of structuring in Na-alginate MRS with MPs depended on the interfacial adhesion among the particles and the polymer. (**a**)—strong adhesion between Na-alginate and MPs leads to the polymeric layers at the surface which favor fluid-type behavior. It is the case of $Fe_3O_4$ and Ni@C MNPs. (**b**)—weak interaction at the interface leads to the networking of MPs due to the magnetic forces which promote elastic deformation and the gelation of MRS. It is the case of Fe and Ni MPs.

It is worthwhile noting that the presented specific behavior of MRS suspensions was observed at a low frequency (1 Hz) of deformation. Low-frequency mode is modeling slow motions in biomimetic materials, for instance, in their application as scaffolds for tissue regeneration or as working bodies in actuators. Meanwhile, there can also be a high-frequency mode for such materials, for example in drug delivery, if the gel drug carrier is activated by ultrasound. In this respect, it might be interesting to examine the mechanical moduli of the magneto-rheological suspensions and the influence of the interfacial adhesion using ultrasound. An example of such an approach to the properties of gel materials with volume phase transition was demonstrated in ref. [32].

## 5. Conclusions

Rheological properties were characterized for the Na-alginate based magnetic suspensions with four different types of embedded magnetic particles (MPs): magnetite ($Fe_3O_4$), metallic iron (Fe), metallic nickel (Ni), and metallic nickel with a deposited carbon layer (Ni@C). These batches of MPs differed in the enthalpy of the interaction of Na-alginate polymer with the surface, as determined by the calorimetry using a thermodynamic cycle. The viscoelastic properties of Na-alginate suspensions showed two definite regimes depending on the concentration of MPs and their chemical nature. At low content of MPs in suspension the regime of deformation was "fluid-type", which meant that the value of the loss modulus was substantially higher than the value of the storage modulus and the shift angle between stress and strain was close to 90°. At low content of MPs, no noticeable influence of the intensity of molecular interaction at the surface on the rheological properties was revealed. The situation changed if the content of MPs in Na-alginate suspensions increased. While the deformation of suspensions with $Fe_3O_4$ particles remained

"fluid-type", the deformation of suspensions with Fe and Ni metal particles switched to the "solid-type" regime which meant that storage modulus became substantially larger than the loss modulus and the shift angle diminished to low values around $10°$, which is typical for the elasticity of gels. A distinct correlation was found between these two types of deformation to the intensity of interaction between Na-alginate and the surface of MPs. In suspensions with negative enthalpy of interaction at the surface, in other words, strong interaction, viscoelastic behavior was "fluid-type". On the contrary, the pronounced elastic deformation was observed in suspensions with positive enthalpy of interaction (poor interaction). Thus, it may be concluded that the elasticity of magnetic suspensions based on Na-alginate is inversely related to the intensity of molecular interactions at the surface of the embedded magnetic particles. Most likely it stems from the formation of the microscopic network of associated magnetic particles which is favored by the poor interaction with alginate.

**Author Contributions:** Conceptualization, A.P.S. and T.V.T.; methodology, E.V.R. and T.V.T.; software, E.V.R.; validation, A.P.S., E.V.R. and T.V.T.; formal analysis, Y.S.Z. and N.M.K.; investigation, Y.S.Z. and N.M.K. resources, I.V.B.; data curation, A.P.S., E.V.R., Y.S.Z. and N.M.K.; writing—original draft preparation, A.P.S.; writing—review and editing, A.P.S., E.V.R., T.V.T. and A.Y.Z.; visualization, A.P.S. and T.V.T.; supervision, A.P.S.; project administration, I.V.B. and A.Y.Z.; funding acquisition, A.Y.Z. All authors have read and agreed to the published version of the manuscript.

**Funding:** This research was funded by Russian Science Foundation, grant number 20-12-00031.

**Institutional Review Board Statement:** Not applicable.

**Informed Consent Statement:** Not applicable.

**Data Availability Statement:** Not applicable.

**Acknowledgments:** Valuable technical assistance of A. I. Medvedev in XRD studies, A. M. Murza-kayev in TEM, and K. G. Balymov in magnetic characterization are highly appreciated.

**Conflicts of Interest:** The authors declare no conflict of interest.

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
