# Peer review of "Gelation in Alginate-Based Magnetic Suspensions Favored by Poor Interaction among Sodium Alginate and Embedded Particles"

_applsci, doi:10.3390/app13074619_

Round 1

Reviewer 1 Report

The manuscript presents an interesting work on the characterizations on the alginate hydrogels with nanoparticles in terms of rheological and calorimetric measurements. The results are presented well in the manuscript. The overall quality of the manuscript is decent. The reviewer recommends acceptance on the manuscript with some additional discussion. The detail comments are listed below.

1.       Oscillatory mode with low oscillational frequency is highly dependent on the sample geometry. Please add some discussion regarding the sample geometry selection.  

2.       Usually, frequency dependent mechanical test is an important part of rheological tests. The frequency dependent is highly dependent on the fabrication processing and the stability of the hydrogels. Please add some discussion regarding this point.

3.       Uniformity is important for producing hydrogels, especially, with nanoparticles. Please add some discussion regarding the uniformity of the particle distribution and mechanical properties contrast.

4.       Ultrasonic characterizations are also potential to be applied in hydrogels for measuring the frequency dependent mechanical properties or energy absorptions. Please add some short introduction with reference: 10.3390/polym12071462. 

Author Response

Dear reviewer,

Thank you for the helpful and friendly comments on our manuscript. All of them resulted in certain additions to the text. Here are the responses to the specific comments.

Response 1

The initial objective of the study was testing of liquid magneto-rheological suspension (MRS) based on alginate water solutions. These suspensions were examined as droplets put in a standard cone-and-plate cell of the Haake MARS rheometer. Hence, no definite geometry of a sample was provided. At high load of magnetic particles we have, however, found that the behavior of the liquid suspensions tended to gelation and the suspensions showed up increasing elasticity. Nevertheless, they still retained restricted fluidity and we were able to keep the initial cone-and-plate setup of the rheometer.  We have added this information to the experimental section

Response 2

At the first step frequency dependence experiments in frequency range 0.1 – 100 Hz at 5 Pa stress were performed for alginate-based MRSs. In this range the dependencies for G’ and G” were linearly increasing in double log scale, with G”>G’ and their cross-over at approximately 80 Hz.  Based on these results the frequency 1 Hz was chosen as the level for the study on the stress-dependent rheological properties for MRSs. Typically, process of gelation in polymer solutions is examined at low frequency of the oscillatory stress and we took 1 Hz as a close value to a lower limit in frequency measurements (0.1 Hz). Information on preliminary frequency measurements has been added to the text

Response 3

Yes, we certainly agree. The alignment of magnetic particles in ferrogel or MRS is an important factor. In our study the suspensions were mixed as it was written in the Experimental section. No special alignment, for instance, by the application of the external magnetic field, was provided. Thus, the distribution of magnetic particles in MRS, presumably, was random. Meanwhile, we understand that the randomness was just macroscopic. Certainly, the magnetic particles have a strong tendency to aggregation and they can aggregate. The examples of such aggregation were shown in our earlier paper ref [26]. Unfortunately, we could not manage to visualize and somehow quantify this aggregation in present study as the content of magnetic particles was too large for the optical microscopy. We have included brief discussion on that in the text.

Response 4

It is a helpful advice, and it is an interesting paper, thank you. We will consider this possibility in future studies. The reference has been added to the list.

Reviewer 2 Report

I read the article Gelation in Alginate-Based Magnetic Suspensions Provided by Poor Interaction among Sodium Alginate and Embedded Particles.

The article is presented very poorly. The title seems very confusing. Abstract is lacking the proper statistical math data. It is difficult to see, where the comparison is carried out to prove whether null hypothesis or alternate hypothesis. citation is poor e.g. a very short sentence (Alginate is one of the natural polymers extensively used in various biomedical applications, healthcare and food products [1–3]) is powered with three references. 

Similarly methodology needs extensive comprehensiveness in order to make it understandable.

Author Response

Dear reviewer,

Thank you for your helpful comments on our manuscript. According to them we have checked again the text and made some changes.

We still do not consider the title of the manuscript as extremely confusing. In our opinion it gives the essence of our results as we see them. Following this comment we have change the word “provided“ to “favored” in the title to emphasize that the marked effect should be regarded not a solid proof but a reasonable explanation.

The manuscript presents a comparative physicochemical study of the rheological and thermodynamic properties of magneto-rheological suspensions and reveals the qualitative correlation among them. No quantitative evaluations were done. In our opinion there was no statistical math to be put in the Abstract. We agree that alginate is one of the most widely studied natural polysaccharides and, certainly, we were able to provide a large list of references concerning the fields of alginate application. However, it was not the objective of the study and we followed an advice of editorial not to extend references which are not directly related to the presented research. Meanwhile, we agree that the cited sentence is too uncertain and we have revised the beginning of the Introduction section and have added references to some books on alginate as well.

The methods of the study were not novel; therefore we made their description brief. Maybe, it appeared to be too brief. We have revised the Experimental section according to this comment.

Reviewer 3 Report

N/A

Author Response

Dear reviewer,

Thank you for reading our manuscript.

There were no comments from your side, and obviously we have nothing to respond to. Meanwhile, we have revised the manuscript according to comments of other two reviewers and you may look at the revisions provided

Round 2

Reviewer 2 Report

The title is still confusing and unable to understand the idea. The authors did not carry the suggested changes. They are again suggested to revise as per previous comments.

Author Response

Dear reviewer,

In the manuscript we present the physicochemical study on the properties of magneto-polymeric suspensions based on alginate solutions with embedded magnetic particles. Any measurable property of the physicochemical system takes its origin in the balance of all molecular interactions of the interior. It is not a hypothesis which needs to be proven, it is the basis of physical consideration. If there are no chemical reactions in the system like in the case of alginate-based MRSs then all the features of the macroscopic properties are governed by the following types of interactions: hydration of alginate, hydration of solid magnetic particle, interfacial adhesion of alginate to the surface of particles, and magnetic forces among particles. The study was focused on the influence of the interfacial adhesion on the rheological properties of alginate MRSs. We have determined the enthalpy of adhesion among alginate polymer and the surface of four types of magnetic particles and have found correlation with the dynamic moduli of MRSs.  This correlation turned out counterintuitive. Gelation was favored not by good adhesion of alginate to the surface but oppositely by the poor adhesion. It is what the paper was about.

To be more certain we have added some general consideration on the importance of molecular interactions to the Discussion section in the first paragraph.

The layout of the paper is typical for polymer physics and chemistry. Methods are usually described as we did it. The accuracy of the measurements is substantially higher that the registered values. The accuracy of calorimetric measurements was given in the original manuscript, but the accuracy of dynamic mechanical measurements was not. We gave it now in the current revision. We have made our best in presenting the results in “physical fashion”. New additions in the text are given in green font.

Apparently you have a different vision on how the data should be presented. We kindly ask you to be more specific on what points in the paper need clarification in your opinion.

Sincerely,

Alexander Safronov